# Management of Nutritional Needs in Pediatric Oncology: A Consensus Statement

**DOI:** 10.3390/cancers14143378

**Published:** 2022-07-11

**Authors:** Francesco Fabozzi, Chiara Maria Trovato, Antonella Diamanti, Angela Mastronuzzi, Marco Zecca, Serena Ilaria Tripodi, Riccardo Masetti, Davide Leardini, Edoardo Muratore, Veronica Barat, Antonella Lezo, Francesco De Lorenzo, Riccardo Caccialanza, Paolo Pedrazzoli

**Affiliations:** 1Department of Hematology/Oncology, Cell Therapy, Gene Therapies and Hemopoietic Transplant, Bambino Gesù Children’s Hospital, 00165 Rome, Italy; francesco.fabozzi@opbg.net (F.F.); angela.mastronuzzi@opbg.net (A.M.); 2Department of Pediatrics, University of Rome Tor Vergata, 00165 Rome, Italy; 3Hepatology Gastroenterology and Nutrition Unit, Bambino Gesù Children Hospital, 00165 Rome, Italy; chiaramaria.trovato@opbg.net; 4Pediatric Hematology-Oncology, Fondazione IRCCS, Policlinico San Matteo, 27100 Pavia, Italy; m.zecca@smatteo.pv.it (M.Z.); s.tripodi@smatteo.pv.it (S.I.T.); 5Pediatric Oncology and Hematology “Lalla Seràgnoli”, IRCCS Azienda Ospedaliero-Universitaria di Bologna, 40138 Bologna, Italy; riccardo.masetti5@unibo.it (R.M.); davide.leardini3@studio.unibo.it (D.L.); edoardo.muratore@studio.unibo.it (E.M.); 6SC Onco-Ematologia Pediatrica, AOU Città della Salute e della Scienza, 10126 Torino, Italy; baratveronica@gmail.com; 7Dietetic and Clinical Nutrition Unit, Children’s Hospital Regina Margherita, AOU Città della Salute e della Scienza, 10126 Turin, Italy; alezo@cittadellasalute.to.it; 8Italian Federation of Volunteer-Based Cancer Organizations, 00187 Rome, Italy; fdelorenzo@favo.it; 9Clinical Nutrition and Dietetics Unit, Fondazione IRCCS Policlinico San Matteo, 27100 Pavia, Italy; r.caccialanza@smatteo.pv.it; 10Medical Oncology Unit, Fondazione IRCCS Policlinico San Matteo and Department of Internal Medicine and Medical Therapy, University of Pavia, 27100 Pavia, Italy; p.pedrazzoli@smatteo.pv.it

**Keywords:** childhood cancer, nutritional support, supportive care

## Abstract

**Simple Summary:**

Nutritional management is an underestimated issue in treating pediatric cancer, since a systematic approach is currently lacking. In this consensus statement, a cohort of 12 experts selected from four different tertiary pediatric oncology centers formulated 21 clinical questions regarding the identification and treatment of nutritional issues in children with cancer. These questions were discussed, and practical recommendations were provided. With this paper, we aimed to give consensus-based guidance for addressing the nutritional needs of children with cancer, filling a gap in the field.

**Abstract:**

Malnutrition, intended as both overnutrition and undernutrition, is a common problem in children with cancer, impacting quality of life as well as survival. In addition, nutritional imbalances during childhood can significantly affect proper growth. Nevertheless, there is currently a lack of a systematic approach to this issue in the pediatric oncology population. To fill this gap, we aimed to provide practice recommendations for the uniform management of nutritional needs in children with cancer. Twenty-one clinical questions addressing evaluation and treatment of nutritional problems in children with cancer were formulated by selected members from four Italian Association of Pediatric Hematology and Oncology (AIEOP) centers and from the Survivorship Care and Nutritional Support Working Group of Alliance Against Cancer. A literature search in PubMed was performed; during two consensus meetings, all recommendations were discussed and finalized using the nominal group technique. Members representing every institution voted on each recommendation. Finally, recommendations were approved by all authors.

## 1. Introduction

The result in treating pediatric cancer is one of the most successful stories in medicine over the last decades, with an overall survival (OS) now exceeding 80% in high income countries [1]. These impressive results are due to the improvement in both antineoplastic therapies and supportive care; regarding the latter, nutrition plays an important role, impacting both short- and long-term outcomes.

The importance of nutritional status in patients with cancer is well established [2]. Moreover, with respect to their adult counterparts, children are at a developmental stage where nutritional imbalances can significantly impact proper growth. Indeed, malnutrition, defined by WHO as both undernutrition and overnutrition, is very common in children with cancer, occurring in up to 70% and between 25% and 75%, respectively [2].

Knowing this, it became evident that a proper nutritional support was essential from diagnosis, during treatment, and even beyond for long-term survival. Nevertheless, standard recommendations for nutritional care are lacking, as pointed out by the clinical practice survey conducted by the Nutrition Committee of the Children’s Oncology Group (COG) [3,4]. The results of this survey shed light on this issue, leading to a growing awareness of its importance in the care of cancer children and, consequently, to a significant increase in scientific publications in this area in recent years [5].

Based on this framework, in this position paper we aimed to provide basic recommendations applicable in high-income country pediatric oncology centers.

## 2. Materials and Methods

This study follows the Revised Standards for Quality Improvement Reporting Excellence (SQUIRE 2.0) [6].

The authors of this paper were selected members from four different Italian Association of Pediatric Hematology and Oncology (AIEOP) centers, including pediatric oncologists and pediatric gastroenterologists with specific skills in nutrition, and from the Survivorship Care and Nutritional Support Working Group (Nutritional Support Commission) of Alliance Against Cancer, which is a National Oncology Network founded in 2002 by the Italian Ministry of Health, currently joined by 28 Italian Institutes. Our aim was to identify nutritional problems in children with cancer and provide basic recommendations applicable in all Italian pediatric oncology institutions and in virtually every pediatric oncology center in high income countries.

A literature search in PubMed was performed with particular reference to the latest five years. The search used combinations of the following terms: cancer, children, childhood cancer, pediatric oncology, nutrition, malnutrition, cachexia, obesity, nutritional practice, nutritional support, and artificial nutrition. Key words were linked using the “OR” Boolean function, and the results of the single components were combined by using the “AND” Boolean function. Guidelines, clinical trials, and observational studies written in English were selected. Due to aim of this paper, the literature review focused on studies conducted in high-income countries, giving less weight to those conducted in low-middle income countries.

Clinical questions relevant for the evaluation and treatment of nutritional issues in children with cancer were formulated by the authors after several rounds of e-discussion from 1 August 2021 to 30 September 2021. These questions were subsequently divided and assigned to different subgroups. Finally, in December 2021 and January 2022, two consensus meetings were held to achieve consensus on and formulate all recommendations. Each recommendation was discussed and modified according to the comments from all participants. Consensus was formally achieved through nominal group technique [7]. Selected members from every institution anonymously voted on each recommendation. A nine-point scale was used (1 = strongly disagree; 9 = fully agree) and votes were reported for each recommendation. Consensus was reached if >75% of the working group members voted >6. All topics received consensus from the expert panel. A consensus was reached for all the questions. The final draft of the recommendations was sent to all authors for approval in February 2022.

## 3. Results

Twenty-one clinical issues were identified and discussed by the expert panel. They are reported here in a Question & Answer form and are summarized in Table 1.

1.Why would nutritional screening be part of supportive care in children with cancer?

Nutritional status during cancer therapy impacts several outcomes, such as overall survival, tolerance to treatment, susceptibility to infection and quality of life, making it a modifiable prognostic factor [5,8]. Consequently, a tailored nutritional support is essential from diagnosis, during treatment, and even beyond long-term survival. Every child with a new cancer diagnosis should be guaranteed a full assessment of their nutritional status, with periodic re-evaluations.

It is also important to add that growth is part of natural history in children; for this reason, the main objective of nutritional treatments is not only to avoid malnutrition, but also to support growth in line with genetic target.

Of note, dietary intake is a main factor modifying gut microbiome, whose composition is associated with various clinical outcomes [9,10].

Votes: 9-8-9-9-9-8-7-9-9-8-9-9-7

2.When should nutritional screening be done in children with cancer?

Assessment of nutritional status using a standardized method should be performed on all patients at diagnosis and repeated periodically throughout the course of treatment, as well as at follow-up.

Patients receiving periods of intensive treatment or at high risk of malnutrition (Table 2) require follow-up every 3–4 weeks. Children receiving less intensive treatment should be evaluated every 3 months, and 6 to 12 monthly intervals while on maintenance phase of treatment, if applicable [11]. Children admitted to intensive care units may need more frequent reassessment.

Votes: 9-8-9-9-8-6-7-9-9-8-8-9-9

3.How should nutritional assessment be done in children with cancer?

Nutritional status should be assessed using a standardized and cost-effective method, as recommended by the Nutrition Working Group (NWG) of the International Society of Pediatric Oncology (SIOP), Committee on Pediatric Oncology in Developing Countries (PODC) [3]. It should consist of:-A. Anthropometric measures.-B. Biochemistry exams.-C. Clinical evaluation.-D. Dietary intake.

Votes: 8-9-9-9-9-9-8-9-9-9-9-8-9

4.What are the anthropometric measures that should be assessed?

The minimal nutritional assessment includes weight, height, body mass index (BMI), and mid-upper arm circumference (MUAC), plotted on WHO growth charts. Although the WHO defines malnutrition using the BMI (±2 Z score), these measurements alone are insufficient for a reliable evaluation in cancer patients; in fact, they can be influenced by the tumor mass, by an imbalance of fluids, and by possible amputations or surgical procedures, leading to an underestimation of malnourished patients [2,11,12,13]. Furthermore, these measurements are not reliable in estimating the body composition of patients, being unable to distinguish fat mass from lean mass; this is an important issue, since cancer patients can undergo a selective loss of lean mass with an imbalance in favor of adipose tissue, even in the presence of a stable weight [5,14,15,16,17,18]. To overcome this hurdle, we recommend at least one methodology able to estimate body composition, such as MUAC.

Tibial length could be useful for children with neurologic impairment. The Z-score determines if the child is stunted, underweight, or acutely malnourished [19,20].

To better assess body composition, Triceps Skinfold Thickness (TSFT), Bioelectrical Impedance Analysis (BIA), air-displacement plethysmography, and Dual energy X-ray Absorptiometry (DXA) could be considered objective and reliable methods that can be compared over time [15].

Votes: 9-8-9-9-8-9-9-9-9-8-9-9-9

5.What are the biochemistry exams that should be performed?

Many biochemical parameters can add information about a patient’s protein status (serum albumin, prealbumin, ans transferrin), organ function (serum urea, creatinine, and liver enzymes), bone health (serum calcium, magnesium, and vitamin D), anemia (iron studies and vitamin levels), evidence of inflammation (serum c-reactive protein (CRP] and erythrocyte sedimentation rate (ESR]) and specific mineral and vitamin deficiencies (zinc and vitamins B12, B1, A, D, and E) [21,22]. However, it should always be considered that these parameters, due to the tumor itself or treatments, can be altered.

More specific laboratory exams, such as retinol binding protein or transferrin receptor dosage, can be used in severely malnourished children to assess malnutrition over time, but they are hardly available in all centers.

Votes: 8-8-9-9-6-8-9-9-9-8-9-7-9

6.What should be investigated during the clinical evaluation?

The clinical evaluation of cancer patients remains essential in order to detect signs of malnutrition, such as the presence of muscle wasting, loss or excess subcutaneous fat, presence of edema, mucous membrane dryness, and hair changes. Clinicians should also consider conditions that may affect oral food intake, such as the inability to chew and swallow, loss of appetite, vomiting, diarrhea, constipation, indigestion, or severe mucositis [11].

Votes: 9-8-9-9-9-9-8-9-9-9-9-9-8

7.What is the role of the dietitian and clinical nutritionist?

A complete dietary history is required for a nutritional assessment, including the intake of macro- and micro-nutrients, food aversions, allergies or intolerances, current eating patterns, family behavior, as well as food hygiene at home [22,23]. Food security can be an important goal to be achieved in special populations, such as nomads or refugees. The full evaluation is best performed by expert personnel, such as dietitians and clinical nutritionists, who should follow the patient during the entire treatment period and after its conclusion. Thus, it would be desirable for every pediatric oncology center to have a clinical nutrition unit/service with expertise in this area. Collaborations between dietitians, clinical nutritionists, and oncologist are pivotal in order to for tailored management of children with cancer.

Votes: 8-8-9-8-8-8-8-9-9-9-8-8-8

8.Can the use of screening tools be useful?

The use of the screening tools could be useful in the global nutritional assessment of children. There are several tools used in pediatrics, such as Simple Pediatric Nutritional Risk Score, to identify children at risk of malnutrition (PNRS) [24]; Screening Tool for Risk of Nutritional Status and Growth (Strong Kids) [25]; Pediatric Yorkill Malnutrition Score (PYMS) [26]; Screening Tool for the Assessment of Malnutrition in Paediatrics (STAMP) [27]; and Nutrition screening tool for childhood cancer (SCAN) [28]. Among these, we suggest using the Screening Tool for Risk of Nutritional Status and Growth (Strong Kids), because it is more balanced and it takes into account many aspects of the disease, the clinical status, and contributing factors, especially related to undernutrition. On the other hand, it views cancer as a single entity without distinguishing the different types or stages of treatment that may have a different impact on nutritional status. Notably, none of these consider malnutrition.

Finally, it would be desirable to create a dedicated score for children with cancer that differentiates by tumor type and expected treatments, as well as taking into account the assessment of overnourishment.

Votes: 7-7-9-9-7-6-7-9-9-7-8-8-7

9.What are the risk factors for malnutrition related to disease and treatment?

When stratifying risk, it is critical to consider the disease and expected treatments. These may carry a high risk of both overnourishment and undernourishment, as summarized by Table 2 [2,11,22,29,30,31,32].

Of note, hypothalamic region has a pivotal role in feeding. Tumors of this region can present with diencephalic syndrome, which is a paradigmatic case of malnutrition: patients have severe weight loss despite a normal appetite and food intake. On the other hand, craniopharyngioma and cranial irradiation can result in obesity by hypothalamic damage [33].

Votes: 9-7-9-9-8-9-9-9-9-9-8-8-8

10.What kind of diet should be suggested?

The requirements of patients with cancer generally correspond with those of children of the same age and sex [11].

Counselling with the family should provide information regarding grocery shopping, food hygiene, food storage, cooking, preparation, and serving, according to the Food and Drug Administration (FDA)-approved food safety guidelines.

To date, the use of a restrictive neutropenic diet has not proven to be superior to regular diets with respect to safe food handling [29,34,35,36,37,38]. It is now clear that restrictive diets have little scientific support and represent an unnecessary burden for the patients and the family, with the risk of further inadequate food intake.

Votes: 8-8-9-9-9-5-6-9-9-6-8-9-9

11.What is the role of “alternative” therapies and diets?

The use of natural health products or special diets in order to fight the disease is very popular among cancer patients [39]. However, there are no high-quality studies demonstrating their effectiveness in achieving pediatric cancer cures [40]. One of the most common strategies is the intake of plant-derived bioactive compounds: if on one side there is no evidence of efficacy of these products, on the other hand, they can potentially interfere with the metabolism of drugs, so their use without medical evaluation should be discouraged [41,42]. The use of special diets, such as the ketogenic diet or calorie restriction, is also to be avoided, since they lack proven efficacy and represent a potential harmful strategy that may negatively interfere on both the efficacy of active treatments, especially in patients at risk of malnutrition or sarcopenia, and the long-term outcome of cancer survivors [42,43].

Votes: 9-8-9-9-9-9-9-9-9-9-9-9-9

12.What is the management for initial starting nutritional support like in children with cancer?

-If a patient is adequately nourished, does not lose weight, and is consuming at least 50% of the recommended nutritional intake, nutritional counselling by an expert dietitian is considered sufficient [44,45,46,47].-Nutritional counselling is mandatory also for overweight and obese patients at diagnosis or during treatment, with special attention for children taking long courses of steroids, who are at risk of sarcopenic obesity (ALL patients).-Nutritional support, starting with oral supplements, is indicated when [48]:○The patient has not high-risk features (Table 2).○The patient is unable to meet the 50% of the daily requirements orally.

Of note, adding nutrient dense regional foods and homemade supplements could be a successful strategy to overcome economic or cultural barriers.

Votes: 9-8-9-9-7-8-7-9-9-8-9-8-7

13.When can enteral nutrition (EN) be considered in children with cancer?

EN should be generally considered in the following conditions:-When the child is unable to take his or her nutritional needs orally (less than 50%) for more than 5 consecutive days; although, a proactive strategy should be preferred in high risk patients, as highlighted by a recent study [48,49].-For severely wasted or malnourished patients, as in low BMI for age (<5th percentile or z score less than –1) or the mid upper arm circumference (MUAC; <5th percentile or z score less than –1).-When the patients have over 5% weight loss since diagnosis, a decrease of >10% in MUAC since diagnosis, or crossing of two growth percentiles over the course of treatment [29,45,50].

Votes: 8-8-9-7-8-9-8-9-9-9-9-8-8

14.Which type of enteral access (nasogastric tube or periendoscopic gastrostomy) is used in children with cancer?

EN may be delivered by several routes: nasogastric, nasoduodenal, and nasojejunal. The first should be the preferred choice, whereas the others should be utilized in patients at risk of pulmonary ingestion or with persistent vomiting.

The nasogastric tube represents the first access route that should be used for EN. Gastrostomy (percutaneous endoscopic, radiologically inserted, or surgical) can be proposed when prolonged support is required (>4–6 weeks) or the nasopharynx needs to be bypassed [22,48,51].

When intragastric feeding is contraindicated, jejunal enteral access could be considered by a nasojejunal tube, jejunal tube introduced through a gastrostomy, or surgical transcutaneous jejunostomy. A jejunal tube needs to be positioned distal to the Treitz ligament to prevent retrograde filling of a dysfunctional stomach.

Votes: 9-8-9-6-8-8-8-9-9-8-9-8-8

15.Which modalities of EN should be used (bolus/continuous) in children with cancer?

EN (by tube, percutaneous endoscopic gastrostomy, or percutaneous endoscopic jejunostomy) can be provided by bolus feeding, continuous feeding, or combination of these two methods. While bolus feeding appears to be more physiological since it tries to mimic normal oral intake, often continuous feeding is safer and better tolerated by the patients [22,48]. We suggest starting with continuous feeding and, if well tolerated (no vomiting or abdominal distension), switching to bolus feeding.

Votes: 9-8-9-5-9-9-9-9-9-8-8-9-9

16.How should an enteral formula be chosen in children with cancer?

The most appropriate nutritional formula should be chosen based on the patient’s age and gastrointestinal function. Standard polymeric formulas are suitable for a functioning gastrointestinal tract, as they contain intact proteins and long-chain triglycerides, whereas formulas containing amino acids and medium-chain triglycerides may be indicated in conditions of malabsorption. In case of fluid restriction or reduced gastric capacity, concentrated formulas can be used, which however are burdened by side effects caused by their osmolarity [22,48].

Votes: 8-8-9-9-9-9-8-9-9-8-9-8-9

17.When should parenteral nutrition (PN) be considered in children with cancer?

When EN is not feasible or inadequate, such as in the presence of severe malabsorption, paralytic ileus, intestinal obstruction, intestinal perforation, short bowel syndrome, intractable vomiting, diarrhea, acute hemorrhage, radiation enteritis, severe adhesions, and intestinal graft versus host disease, PN should be considered [29,48,50]. PN is usually initiated if inadequate EN is expected for at least 5–7 days [22,48].

Votes: 8-8-9-9-8-8-9-9-9-8-9-9-8

18.How personalized should PN be in children with cancer?

PN formulations should be prescribed, taking into account age requirements and nutritional status, as well as the fluid requirement and type of venous access, as central venous access is often required to meet nutrient needs [22]. The metabolic changes in children with cancer may influence individual needs [52].

Minimal enteral feeding should be introduced as soon as the clinical conditions allow because it helps restoring a healthy microbiome, whose role seems essential for intestinal function and global health maintenance [53].

Votes: 8-8-9-9-9-8-9-9-9-9-9-8-9

19.What are the risks related to PN?

The possible complications related to the use of PN are: mechanical or equipment-related complications, such as CVC thrombosis, break, occlusion, or dislodgement; infective complications, for example, CVC-associated infections; and metabolic complications, such as deficiency or excess of PN components (hypertriglyceridemia and hyperglycemia), acid-base or electrolyte imbalance, drug interaction or compatibility problems, intestinal failure associated liver disease, and refeeding syndrome [52]. Gut microbiome dysbiosis can occur with exclusive PN.

Votes: 8-8-9-9-9-8-9-9-9-8-9-9-9

20.When should nutritional assessment be performed in cancer survivors?

In cancer survivors, a regular assessment is highly recommended, based on nutritional risk: monthly for undernourished patients, quarterly for obese, whereas patients without risk factors can be evaluated every six months during first year of follow-up and then yearly [11]. Well-nourished patients with nutritional risk factors (dyslipidemia, hyperglycemia, inadequate eating habits, etc.) should be assessed quarterly in the first year, every six months until the fifth year, and then annually [11].

Votes: 7-8-9-9-8-8-8-9-9-9-8-9-8

21.What are the nutritional risks in cancer survivors?

Survivors of childhood cancer have an increased risk of developing obesity, dyslipidemia, hyperglycemia, metabolic syndrome, osteopenia, and osteoporosis [54,55,56,57]. Additionally, other nutritional risk factors, such as inadequate eating habits, smoking, sedentary lifestyle, and alcoholism may increase the already higher cardio-vascular risk of these patients [11].

Votes: 7-8-9-9-9-9-8-9-9-9-9-8-9

## 4. Discussion

The importance of nutritional support in cancer patients has recently been highlighted in several publications [58,59]. This is particularly significant in childhood, when malnutrition, both in excess and in deficiency, can influence not only treatment outcomes, but also the correct development of the child. This problem has recently been highlighted especially in developing countries, where more than 80% of pediatric cancer cases are diagnosed, and malnutrition and socioeconomic disadvantages are especially prevalent even more in normal population [60]. Considering all these, recent efforts by the Nutrition Working Group (NWG) of the International Society of Pediatric Oncology (SIOP) have resulted in a framework for assessment and optimal delivery of nutritional care in low-middle income countries [3]. However, malnutrition among children with cancer represents a real hurdle also in high-income countries, but similar recommendations are currently lacking, indicative of an underestimation of this problem.

Literature on nutritional issues with children with cancer is an ongoing and intriguing area of research that is represented by gut microbiome. Recent findings have shown that gut microbiome composition may have a significant role in impacting outcomes in several diseases, including pediatric cancer [61,62]. Consequently, its modification and/or restoration by nutritional interventions, such as limiting lactose intake, increasing fiber intake, as well as avoiding exclusive PN, may represent an important target to ameliorate cancer care [63,64].

Of note, the growing attention from the media on hypocaloric anti-cancer diets has created a background of speculations and ambiguous messages. Such diets are not recommended as they could worsen protein-calorie intake and negatively interfere with cancer care, particularly in malnourished patients and those at nutritional risk.

## 5. Limitations

Several important limitations of this study should be underlined. Primarily, the paucity of clinical studies makes our recommendations based primarily on expert opinion-level evidence. This calls for the need to expand clinical research on this pivotal topic, to make it possible to formulate evidence-based guidelines. In addition, our recommendations refer to a panoramic view of pediatric oncology, without focusing on specific areas such as bone marrow transplantation and cellular therapies that need dedicated indications. Another important limitation is that our recommendations do not adequately address the needs of special populations, such as nomadic populations or refugees. Important barriers, both economic and cultural, must be taken in account, especially when a fairly invasive intervention is proposed.

Finally, the panel of experts was composed of physicians all from one country (Italy), and this could reduce the generalizability of our results to other high-income countries.

## 6. Conclusions

This consensus statement provides practice recommendations addressing nutritional care in pediatric oncology, aiming to fill an important gap in this field. Based on the importance of nutritional state as an independent modifiable prognostic factor, it is desirable that the present work, by highlighting this important issue, will facilitate the initiation of multicentric clinical trials to confirm, refine, or even modify these recommendations.

## Figures and Tables

**Table 1 cancers-14-03378-t001:** Summary of recommendations.

Questions	Recommendations
Why would nutritional screening be part of supportive care in children with cancer?	The main objective of nutritional treatments is not only to avoid malnutrition, but also to support growth in line with genetic target.
When should nutritional assessment be done in children with cancer?	Assessment of nutritional status should be performed on all patients at diagnosis and repeated periodically during treatment and follow-up.
How should nutritional screening be done in children with cancer?	A-B-C-D methods could be considered a useful method for nutritional screening in children with cancer.
What are the anthropometric measures that should be assessed?	Weight, height, body mass index (BMI), and mid-upper arm circumference (MUAC) plotted on WHO growth charts could be considered part of a minimal nutritional screening.
What are the biochemistry exams that should be performed?	Biochemical exam should include protein status, organ function, bone health, anemia, evidence of inflammation, and specific mineral and vitamin deficiencies.
What should be investigated during the clinical evaluation?	Clinical evaluation should detect signs of malnutrition and consider conditions that may affect oral food intake.
What is the role of the dietitian and clinical nutritionist?	Collaboration between dietitians, clinical nutritionists, and oncologist is pivotal.
Can the use of screening tools be useful?	Screening Tool for Risk of Nutritional Status and Growth (Strong Kids) seem to be balanced and takes into account several aspects.
Which risk factors for malnutrition are related to disease and treatment?	Some specific tumors and some specific therapies are more at risk of both overnourishment and undernourishment.
What kind of diet should be suggested?	A diet corresponding to those of children of the same age and sex should be proposed. Counselling on grocery shopping, food hygiene, food storage, cooking, preparation, and serving according to the FDA-approved food safety guidelines should be carried out to families.
What is the role of “alternative” therapies and diets?	There are no high-quality studies demonstrating the effectiveness of natural health products or special diets in pediatric cancer cures.
What is the management for initial starting nutritional support like in children with cancer?	Nutritional support, starting with oral supplements, is indicated when the patient has no high-risk features or when they are unable to meet the 50% of the daily requirements orally. If a patient is adequately nourished, does not lose weight, and is consuming at least 50% of the recommended nutritional intake, nutritional counselling by an expert dietician is considered sufficient. Nutritional counselling is mandatory also for overweight and obese patients at diagnosis or during treatment, with special attention to children taking long course of steroids, who are at risk of sarcopenic obesity (ALL patients).
When can enteral nutrition (EN) be considered in children with cancer?	- When the child is unable to take his or her nutritional needs orally (less than 50%) for more than 5 consecutive days. - For severely wasted or malnourished patients (BMI for age <5th percentile or z score less than –1) or MUAC (<5th percentile or z score less than –1). - When the patients have over 5% weight loss since diagnosis; a decrease of >10% in MUAC.
Which type of enteral access (nasogastric tube or periendoscopic gastrostomy) is used in children with cancer?	- Nasogastric tube is the first access that should be used. - Gastrostomy can be proposed when prolonged support is required (>4–6 weeks) or the nasopharynx needs to be bypassed. - Jejunal enteral access could be considered when intragastric feeding is contraindicated.
Which modalities of EN should be used (bolus/continuous) in children with cancer?	We suggest starting with continuous feeding and, if well tolerated (no vomiting or abdominal distension), switching to bolus feeding.
How should an enteral formula be chosen in children with cancer?	- Standard polymeric formulas are suitable for a functioning gastrointestinal tract. - Formulas containing amino acids and medium-chain triglycerides may be indicated in conditions of malabsorption. - Concentrated formulas can be used in case of fluid restriction or reduced gastric capacity.
When should a parenteral nutrition (PN) be considered in children with cancer?	PN should be considered when enteral nutrition is not feasible or inadequate.
How personalized should PN be in children with cancer?	PN formulations should be prescribed, taking into account age requirements, nutritional status, fluid requirement, and type of venous access.
What are the risks related to PN?	The possible complications related to the use of PN are mechanical or equipment-related complications, infections and metabolic complications, acid-base or electrolyte imbalance, drug interaction, intestinal failure associated liver disease, and refeeding syndrome.
When should nutritional assessment be done in cancer survivors?	Nutritional assessment in cancer survivors should be done during the first year of follow-up: Monthly for undernourished patients; Quarterly for obese children and well-nourished patients with nutritional risk factors; Six months for children without risk factors

**Table 2 cancers-14-03378-t002:** Factors identifying a high risk for undernourishment and overnourishment.

High Risk Factors for Undernourishment	High Risk Factors for Overnourishment
Solid tumors with advanced stages at diagnosis	Total body or abdominal or cranial irradiation
Ewing sarcoma	Craniopharyngioma
Medulloblastoma and other high grade brain tumors	Administration of prolonged corticosteroid therapy or other drugs increasing body fat stores
Diencephalic tumors	
Head and neck tumors	
Age < 2 months	
Relapsed disease	
Administration of highly emetogenic regimens	
Administration of regimens associated with severe gastrointestinal complications, such as constipation, diarrhea, loss of appetite, mucositis, or enterocolitis	
Administration of radiation to the oropharynx, esophagus, or abdomen	
Post-surgical complications, such as prolonged ileus or short gut syndrome	
Stem cell transplantation with myeloablative conditioning regimens

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
