# Peer review of "Management of Nutritional Needs in Pediatric Oncology: A Consensus Statement"

_cancers, 2022, doi:10.3390/cancers14143378_

Round 1
Reviewer 1 Report
This manuscript is an interesting approach into the pediatric and nutritional aspect of cancer in children. It could be the start of many clinical trials that could target the needs in this population. However, the discussion can be more detailed and expanded. Comparing different trials or similar studies in this manuscript, it seems very generalized.
What results of the database search was used to backup this manuscript? What countries or populations were involved?
Also, it will be interesting to include nutritional experts into the physicians so it could have more weight into the nutritional part of the study that seems the most important.
Here are some corrections and suggestions:
Table 1 Recommendations is misspelled.
Table 1 Questions 1,2,3,9,11 and 20 are missing a period.
Table 2 missing periods in sentences.
Suggestion line 161-162 : "However, it should be always considered that these parameters can be altered due to the tumor itself or treatments. " instead put : " However, always should be considered that these parameters due to the tumor itself or treatments can be altered"
Line 184 is pivotal to are pivotal.
217 instead of food shopping use grocery shopping
Line 341 issues* with*
Line 342 that is represented*
Author Response
Dear Editor, Dear Reviewer,
We are very grateful for your valuable suggestions to improve our manuscript.
Please find below the requested clarifications:
- The discussion can be more detailed and expanded. Comparing different trials or similar studies in this manuscript, it seems very generalized.
Thank you very much for the suggestion. We have developed the discussion more.
- What results of the database search was used to backup this manuscript? What countries or populations were involved?
Thank you very much for the wise consideration. We added a period in the methods section to specify that we favored studies conducted in high-income countries
- Also, it will be interesting to include nutritional experts into the physicians so it could have more weight into the nutritional part of the study that seems the most important.
Thank you very much for the valuable suggestion. Our aim is to involve more clinical nutritionists for future work in this field.
- Here are some corrections and suggestions: Table 1 Recommendations is misspelled. Table 1 Questions 1,2,3,9,11 and 20 are missing a period. Table 2 missing periods in sentences. Suggestion line 161-162 : "However, it should be always considered that these parameters can be altered due to the tumor itself or treatments. " instead put : " However, always should be considered that these parameters due to the tumor itself or treatments can be altered". Line 184 is pivotal to are pivotal. 217 instead of food shopping use grocery shopping. Line 341 issues* with*.Line 342 that is represented*
We are very grateful to the reviewer for highlighting these errors. We corrected them as suggested in the final draft of the manuscript.
We look forward to hearing from you at your earliest convenience, and we thank you very much for your kind attention and important recommendations.
Best regards,
Francesco Fabozzi on behalf of all authors
Reviewer 2 Report
Congratulations to the authors for this interesting work. The methodological section should be implemented with a more detailed description of the research strategies. The final number of works that have been taken into consideration and the reasons for any exclusions are not indicated. I recommend adding a flowchart that correctly describes your decision making process.
I recommend, as a future development of the work, to extend the consensus in an extra national context, for example in the context of an international scientific society.
Author Response
Dear Editor, Dear Reviewer,
We are very grateful for your valuable suggestions to improve our manuscript.
Please find below the requested clarifications:
- The methodological section should be implemented with a more detailed description of the research strategies. The final number of works that have been taken into consideration and the reasons for any exclusions are not indicated. I recommend adding a flowchart that correctly describes your decision making process.
We thank the reviewer very much for this important consideration. Our work was not designed as a systematic or narrative review of the literature but rather as a state of art of current literature. Analysis of these evidences led us to make recommendations based on expert opinion. Based on these premises, we not provide a degree of evidence. However, we have added some sentences to better clarify the selection criteria for the studies we considered to support the recommendations.
We look forward to hearing from you at your earliest convenience, and we thank you very much for your kind attention and important recommendations.
Best regards,
Francesco Fabozzi on behalf of all authors
Round 2
Reviewer 1 Report
The introduction can be more elaborate, since seems very concentrated and lacks more explanatory details for the reader that is not an expert on the field.
It seems that the nutritional part of the study is of most importance, I suggest for future research that the nutritionist expertise input be included for the manuscript to have more credibility and substance.
The corrections were made in previews reviews.
The manuscript has potential of polishing, the English needs to be revised, and more details as stipulated on my top review in order for people that are not expert in this matter can grasp the knowledge.
Author Response
Dear Editor, Dear Reviewer,
We are very grateful for your valuable suggestions to improve our manuscript.
Please find below the requested clarifications:
- The introduction can be more elaborate, since seems very concentrated and lacks more explanatory details for the reader that is not an expert on the field.
Thank you very much for the suggestion. We have developed the introduction more, trying to be more understandable even for those who are not experts in this field.
- It seems that the nutritional part of the study is of most importance, I suggest for future research that the nutritionist expertise input be included for the manuscript to have more credibility and substance.
Thank you very much for the wise consideration. Of all the authors, four (namely CM Trovato, A Diamanti, A Lezo, and R Caccialanza) are clinical nutritionists with expertise regarding nutrition in patients with cancer. Our intention for future research is to further expand the board of nutritionists
- The manuscript has potential of polishing, the English needs to be revised, and more details as stipulated on my top review in order for people that are not expert in this matter can grasp the knowledge.
We are very grateful to the reviewer for the suggestions. English was revised, and more details were provided to explain better our recommendations.
We look forward to hearing from you at your earliest convenience, and we thank you very much for your kind attention and important recommendations.
Best regards,
Francesco Fabozzi on behalf of all authors
